# Assessing COVID-19 pandemic's impact on essential diabetes care in Manila, the Philippines: A mixed methods study

**Greco Mark B. Malijan**[1], **John Jefferson V. Besa**[2,3], **Jhaki Mendoza**[2], **Elenore Judy B. Uy**[4], **Lijing L. Yan**[5,6]*, **Truls Østbye**[5,7], **Lia Palileo-Villanueva**[2,3]

**1** San Lazaro Hospital–Nagasaki University Collaborative Research Office, Manila, Philippines, **2** College of Medicine, University of the Philippines Manila, Manila, Philippines, **3** Department of Medicine, Philippine General Hospital, University of the Philippines Manila, Manila, Philippines, **4** Asia Pacific Center for Evidence-Based Healthcare, Manila, Philippines, **5** Global Health Research Center, Duke Kunshan University, Jiangsu Province, China, **6** School of Public Health, Wuhan University, Wuhan, China, **7** Department of Family Medicine and Community Health, Duke University, Durham, North Carolina, United States of America

* lijing.yan@duke.edu

**Data Availability Statement:** The data supporting the findings of this study are provided as part of the supplementary information.

## Abstract

The COVID-19 pandemic directly increased mortality and morbidity globally. In addition, it has had extensive indirect ill effects on healthcare service delivery across health systems worldwide. We aimed to describe how patient access to diabetes care was affected by the pandemic in Manila, the Philippines. We used an explanatory, sequential mixed method approach including a cross-sectional survey (n = 150) and in-depth interviews of patients (n = 19), focus group discussions of healthcare workers (n = 22), and key informant interviews of health facility administrators (n = 3) from October 2021 to January 2022. Larger proportions of patients reported absence of livelihood (67.3%), being in the lowest average monthly household income group (17.3%), and disruptions in diabetes care (54.0%) during the pandemic. They identified the imposition of lockdowns, *covidization* of the healthcare system, and financial instability as contributors to the reduced availability, accessibility, and affordability of diabetes-related consultations, medications, and diagnostics. At least a quarter of the patients experienced catastrophic health expenditures across all areas of diabetes care during the pandemic. Most healthcare workers and administrators identified telemedicine as a potential but incomplete tool for reaching more patients, especially those deemed lost to follow-up. In the Philippines, the pandemic negatively impacted access to essential diabetes care.

## Introduction

As early as March 2020, the World Health Organization (WHO) cautioned governments of the ill effects of the coronavirus disease (COVID-19) pandemic on overwhelmed health systems, causing dramatic rise in direct mortality from the outbreak and indirect morality from preventable and treatable conditions in most countries [1]. Communicable disease prevention,

**Funding:** This work was supported by the Duke Global Health Institute, SingHealth Duke-National University of Singapore Global Health Institute, and Duke Kunshan University under the collaborative scheme "Health Systems Strengthening to Better Prepare for and Respond to Pandemics" (through LLY and TØ). The funders had no role in study design, data collection and analysis, decision to publish, or preparation of the manuscript.

**Competing interests:** The authors have declared that no competing interests exist.

reproductive health services, core services for vulnerable populations such as infants and older adults, provision of medications for chronic diseases, critical facility-based therapies, and auxiliary services were identified as high-priority areas to avert indirect morbidities from the pandemic [2]. Despite these early warnings, most health systems worldwide struggled to cope.

Non-communicable disease (NCD) care is increasingly recognized as an important indicator of health system resilience [3]. The WHO reported that up to 75% of member states experienced considerable disruptions to NCD services attributed to the pandemic across all regions and income groups by September 2020 [4]. Human resources allocated to NCD programming either shifted their entire work to COVID-19 services or handled additional responsibilities related to pandemic response. Rehabilitation, hypertension and diabetes care, asthma care, palliative care, and urgent dental care were the services deemed most likely to be disrupted. Moreover, studies on the impact of COVID-19 on health services in the WHO Southeast Asia region found significant reductions in cancer-related outpatient services, cardiovascular admissions, and respiratory illness diagnoses [5]. Alarmingly, individuals with NCDs such as diabetes, cardiovascular diseases, and cancer are at greater risk of morbidity and mortality from COVID-19 compared to individuals without these conditions [6].

The Philippine government's securitized response to the pandemic was characterized by sustained border and mobility restrictions, stringent penalties against erring individuals, and rapid realignment of health financing and services that reflected the view of COVID-19 being an existential threat [7,8]. These had multifaceted impact on the health system. A systematic literature review of policies and a qualitative study involving system managers in early 2021 found that community quarantines universally limited health service access and delivery, with greater cascading effects on those needing specialized care [9]. Furthermore, analyses of claims data from PhilHealth, the national health insurance system, found that inpatient care for diabetes, cancer, stroke, and chronic kidney disease had dropped by 57% in 2020 compared to the previous year [10] and that the poorest indigent members experienced the greatest reductions in care [11]. The analyses also found that access limitations were most evident during the implementation of the most stringent lockdown policies.

Levesque and collaborators view access as "the opportunity to reach and obtain appropriate health care services in situations of perceived need for care" [12]. Healthcare access is conceptualized as the interface between the characteristics of providers and health systems and the characteristics of individuals and populations. On the one hand, for optimal healthcare access, services must be approachable, acceptable, available, affordable, and appropriate. On the other hand, individuals must have the ability to perceive, to seek, to reach, to pay, and to engage with existing systems. This framework allows operationalization of the various processes involved in accessing patient-centered care. It offers a lens through which to study both the direct and indirect impact of the pandemic.

Because almost 90% of persons living with diabetes have at least one NCD comorbidity, diabetes serves as an important tracer condition in studying NCD burden [13] and has been used to evaluate health systems and primary care [14,15]. A survey of healthcare professionals from 47 countries found that diabetes care was deemed the primary care service most likely to be affected by the pandemic [16]. Comprehensive diabetes care involves regular interactions with various health service providers at different healthcare levels to facilitate the diagnosis, monitoring, therapy, and rehabilitation and to prevent complications [17]. The difficulties in accessing diabetes care during the pandemic may be a product of delayed care seeking, limited self-care practice, transportation difficulties, outpatient clinic lockdowns, decreased inpatient capacity, and staff and medicine shortages [18].

We aimed to describe how access to diabetes care, as a tracer condition for NCDs in general, was affected by the COVID-19 pandemic in Manila, the Philippines. Specifically, we

assessed the availability, accessibility, and affordability of consultations, medications, and diagnostic procedures associated with diabetes primary care through a mixed methods study.

## Materials and methods

### Study design

We used an explanatory, sequential mixed method approach to assess the impact of COVID-19 on diabetes primary care. We conducted a cross-sectional survey followed by in-depth interviews (IDIs) among patients with diabetes. To corroborate the quantitative and qualitative results from the patient perspective, we conducted focus group discussions (FGDs) and key informant interviews (KIIs) among healthcare providers and health facility administrators. We deemed the mixed method approach to be most appropriate in achieving the overall study aim as it allows triangulation of methodology and perspectives that may better capture co-created realities.

### Study setting

We recruited patient participants from the Prospective Urban Rural Epidemiological (PURE) study and the outpatient department (OPD) of the Philippine General Hospital (PGH).

The PURE study is a large longitudinal epidemiological study examining the relationship of societal influences on lifestyle behaviors, cardiovascular risk factors, and incidence of NCDs [19]. It has enrolled more than 250,000 participants from more than 25 countries globally. In the Philippines, 2,264 participants residing in contiguous communities from Barangays 178 to 186 in Tondo, Manila, are enrolled in the PURE study as part of its urban cohort. The participants mostly seek consultations at the nearby local health centers, which are public primary care facilities that serve an identified catchment area. Each health center is typically staffed by a team of doctor(s), dentist(s), nurses, midwives, and community health workers (CHWs) [20]. Health services such as consultations, selected medications, including those for hypertension and diabetes, diagnostic tests, and health education and promotion are provided free at point of care regardless of household income. The local health center also serves as the implementing entity for health programs. However, medication stock outs and lack of skilled personnel are challenges faced by these facilities, and the availability of different medication types (e.g., oral hypoglycemic agents, insulin) may vary depending on the resources of the local government unit.

PGH is one of the largest public hospitals in the country and serves as a national tertiary referral center. Located in the City of Manila, its OPD caters to more than 600,000 patients annually. The General Internal Medicine Clinic (GIMC) serves as a typical first point of contact for adult patients seeking care at the OPD. The clinic is staffed by a team of internal medicine resident trainees, nurses, and medical students. In response to the COVID-19 pandemic, PGH implemented a hospital-wide electronic medical record system and teleconsultation services at the OPD.

### Participants

For the cross-sectional survey and IDIs with patients, we used the following eligibility criteria.

### Inclusion criteria

1. Adult 18 years or older

2. Diagnosed with diabetes prior to January 2019 (at least one year before the first report of COVID-19 in the Philippines) by a health professional defined as:

- Patient self-report of physician-adjudicated diagnosis and/or taking any medication for diabetes OR

- Medical records indicating diagnosis of diabetes

3. Participant of the PURE study OR patient consulting at the PGH GIMC through its telemedicine services

## Exclusion criteria

1. Unable to give informed consent (e.g., severe illness, frailty, dementia, psychological or mental incapacity)

2. Seeking care at a specialty clinic for diabetes care such as the PGH endocrine specialty clinic (n.b., patients referred to the specialty clinic are more likely to have more complex medical conditions necessitating specialized care beyond the level of primary care)

Moreover, we recruited hospital administrators, nurses, internal medicine residents, and community health workers for the qualitative component of the study.

## Quantitative data collection

**Recruitment.** Recruitment processes for the cross-sectional study differed between the community and hospital settings, taking into consideration resource limitations and mobility restrictions imposed at the time. We sought permission from the PURE study team to access the list of participants with diabetes in the urban community cohort. We used stratified random sampling, with sex as the stratum-defining variable. A list of ID numbers of eligible participants was used as the sampling frame. Each ID number was assigned using the random number generator function of Microsoft Excel [21]. The assigned random numbers were arranged in ascending order, and participants were recruited consecutively until the desired sample size was reached. To recruit from PGH OPD, we sampled by convenience. We secured a list of patients who were scheduled for follow-up telemedicine consult from October 2021 to January 2022. We called patients consecutively (i.e., earliest scheduled for a particular day was called first) and recruited those eligible until the desired sample size was reached. We also recruited eligible patients referred to the study by their attending physicians.

Verbal informed consent was discussed and secured from eligible participants. For every consenting patient, an informed consent form was accomplished, indicating the participant's name, date of consent, and name and signature of study personnel who obtained consent. Moreover, a screening log was maintained containing participant name and their decision regarding consent to participate. Once having provided consent, study participants were invited for a telephone, online, or in-person interview, depending on their preference. For in-person interviews, the trained research assistants also obtained a written informed consent prior to the administration of the questionnaire. The study team adhered to minimum health and safety protocols based on prevailing guidelines at the time. The informed consent process for participants enrolled at the PGH GIMC further included a review of medical records to secure accurate information on diagnosis, duration of diabetes, latest laboratory results, and list of prescribed medications.

**Instruments.** We used a researcher-administered survey questionnaire (S1 Appendix) to characterize the patients' experience of diabetes care before and during the pandemic. The

questionnaire was administered once, and participants were asked to recall and report their experiences before and during the pandemic. We operationally defined 'pre-pandemic' as the period prior to January 2020 and 'during the pandemic' as the period in the preceding six months of the study period. Adapting the Levesque framework [12], we asked questions related to demographic and diabetes clinical characteristics at the time of enrollment and to the availability, accessibility, and affordability of healthcare consultations, medications, and laboratory monitoring pre-pandemic and during the pandemic. We operationally defined *availability* as the presence of an identified provider for a particular service, *accessibility* as the physical ability of the person living with diabetes to reach a particular service, and *affordability* as the presence of financial means to avail of the service. We considered catastrophic health expenditure, which we defined as foregoing other necessities or borrowing money to afford a component of diabetes care, as an important potential consequence of the lack of affordability and evaluated it in the questionnaire.

Survey data were directly encoded into electronic data collection forms that had undergone pre-testing. We used password-protected devices exclusively for the study. Data were de-identified during data collection. The authors had no access to information that could identify individual participants. Each participant was assigned a unique study ID, and patient identifiers were stored separately from study participant data. We adhered to existing Philippine data privacy laws and research data management guidelines in the conduct of the study.

**Sample size.** We assumed that 50% of individuals with diabetes in the population would report to not have access to DM care during the pandemic. We estimated a population size of 162, assuming an 8% prevalence of diabetes among the 2,264 participants in the PURE community cohort, based on the national prevalence of high fasting blood glucose among adults residing in urban areas [22]. We assessed that the study would require a sample size of 124 for estimating the expected proportion with 5% absolute precision and 95% confidence [23].

## Qualitative data collection

**Instruments.** The qualitative component consisted of interviews among patients and healthcare administrators as well as FGDs among healthcare workers. Semi-structured interview and FGD topic guides (S2 Appendix) were developed to contain key questions tailored for each participant group. The IDIs were intended to further explore patients' perspective on diabetes care and management during the COVID-19 pandemic while the FGDs sought to capture the experiences of healthcare workers in providing diabetes care. Interviews among healthcare administrators were also conducted, focusing on institutional adaptations in delivering diabetes-related healthcare services during the COVID-19 pandemic. Two of the investigators (JM and EM) conducted the IDIs and FGDs using the local language (Filipino) to build better rapport and to address limitations of translations.

**Researcher characteristics and reflexivity.** JM and EM are Filipino- and English-speaking local researchers with background in social sciences and experience in doing qualitative health research in the Philippines. Their background provided a non-medical point of view for the study which the study team deemed advantageous as JM and EM prioritized the lived experiences of accessing and delivering care for DM from varying points of view of study participants. This contrasts with the clinical and epidemiological background the rest of the study team provided.

**Study size and recruitment.** After the quantitative survey, patients were asked about their willingness to participate further in the qualitative component of the study. Those who expressed interest were screened for inclusion into the IDIs using maximum variation of age distribution, sex, and location. The ideal sample size initially identified for the IDIs was 30

based on earlier literature [24,25]. However, data saturation [26], specifically thematic saturation, where no new concepts were emerging relative to our study population [27], was reached after 15 interviews.

KIIs with health administrators and FGDs among internal medicine residents and nurses were completed at the hospital outpatient clinic. Meanwhile, one FGD with CHWs and an interview with an administrator were conducted in one of the PURE study sites in the City of Manila. In PGH, we purposively selected potential participants to achieve representation based on the type of healthcare provider (e.g., resident doctor, nurse, social worker) and sex difference. Similar criteria were used to recruit from the health center, with additional consideration for staff availability.

For all qualitative study participants, verbal informed consent was discussed and audio recorded before the start of any interview or FGD.

## Quantitative data analysis

We summarized demographic and clinical characteristics into proportions and used median [interquartile range] to summarize non-normally distributed data. We collapsed Likert scales that were used to characterize availability, accessibility, and affordability of consultations, medications, and laboratory monitoring. The Likert scale had a range of 1 as very difficult and 10 as very easy, and we categorized any value less than or equal to 8 to be problematic. To assess differences in socioeconomic status, diabetes care, and challenges in access to diabetes care pre-pandemic and during COVID-19, we used McNemar's test for paired or dependent nominal data. In assessing catastrophic health expenditure, we considered those who had no consultation, medication, or laboratory testing in the past six months as separate layers. Hence, we used Chi-squared test to assess the differences in the reported catastrophic health expenditure pre-pandemic and during pandemic. All statistical tests were conducted using 95% confidence level. We performed all statistical tests and visualizations in R version 4.1.2 and RStudio version 2022.12.0+353 [28].

## Qualitative data analysis processes

Two investigators (JM and EM) conducted the preliminary analysis. Patient interviews and KIIs lasted around 45–60 minutes while the FGDs lasted about 60–90 minutes. All recordings were transcribed verbatim. The first analytical process involved inductively coding and thematically analyzing all the transcripts written in the local language to identify emerging patterns. The emerging themes were interpreted relative to our conceptual framework to ensure alignment with the study objectives. This step sought to balance the *emic* (participants' perspective) and *etic* (researchers' perspective) standpoints ensuring reflexivity. The second and last analytical process sought to consolidate the emerging themes with the preliminary quantitative findings through a series of discussion among all investigators.

## Mixed-methods triangulation

In accordance with the explanatory sequential typology, the results of the quantitative component partly informed the analysis of the qualitative component. Data triangulation took place at the point of completion of the statistical analysis of the patient cross-sectional survey and during the interpretation of the interviews and FGDs. We compared and contrasted key findings from both qualitative and quantitative data to arrive at points of convergences and divergences. Thereafter, a second round of coding and qualitative analysis using a deductive approach was conducted by two of the researchers (JM and EM) to further validate our key findings.

This pragmatic approach allowed for the qualitative data and its analysis to add depth to the findings of the quantitative data analysis [29]. The final analysis and interpretation of the sequentially collected, integrated data were agreed upon by the whole study team through consensus building meetings.

### Reporting

We combined and applied key components of the mixed methods research manuscript checklist by Lee and collaborators [30], the Strengthening the Reporting of Observational Studies in Epidemiology guidelines [31] (S1 Checklist), and the Standards for Reporting Qualitative Research guidelines [32] (S2 Checklist) in preparing this manuscript.

### Ethics statement

The study protocol, study procedures including verbal consent process, and all study materials including consent forms, data collection tools, and interview and FGD guides were reviewed and approved by the University of the Philippines Manila–Research Ethics Board (UPMREB Code: 2021-375-01).

## Results

Between October 2021 and January 2022, we recruited and surveyed a total of 100 participants from the PURE community and 50 participants from the PGH OPD (Fig 1). Additionally, 19 patient interviews were conducted from this mix. One interview with health center administrator and one FGD of four CHWs were conducted in the PURE community. Finally, two FGDs involving 13 medical residents, one FGD with five OPD nurses, and two interviews with administrators were conducted in PGH.

### Cross-sectional survey findings

Most of the participants enrolled in the study were middle-aged, female, married, and had completed at least secondary school education (Table 1). The median duration of diabetes diagnosis was 10 years, and the majority had at least one comorbidity, with hypertension being the most frequent.

Before and during the pandemic, membership in any health insurance scheme and in PhilHealth, the main implementing mechanism for universal health coverage [33], was consistently high (Table 2). However, the proportion of patients with any means of livelihood during the pandemic was significantly lower, and the proportion of patients below the PHP 5,000 monthly household income group during the pandemic were significantly higher than prior to the pandemic. There was no significant difference in household size between periods.

There were significant differences in the diabetes care received by the patients between time periods (Table 3). Prior to the pandemic, most patients had been followed in the same healthcare facility for more than two years. However, during the pandemic, more than half of the participants reported having been followed in their healthcare facility for less than a year. Furthermore, 6.7% had to stop taking their medications completely. There was a significant reduction in glycemic control during the pandemic, and almost a quarter of participants did not have any fasting blood sugar (FBS) monitoring.

The proportion of participants who reported challenges in access to consultations, medications, and diagnostic testing during the pandemic increased by up to twice compared to prepandemic (Fig 2).

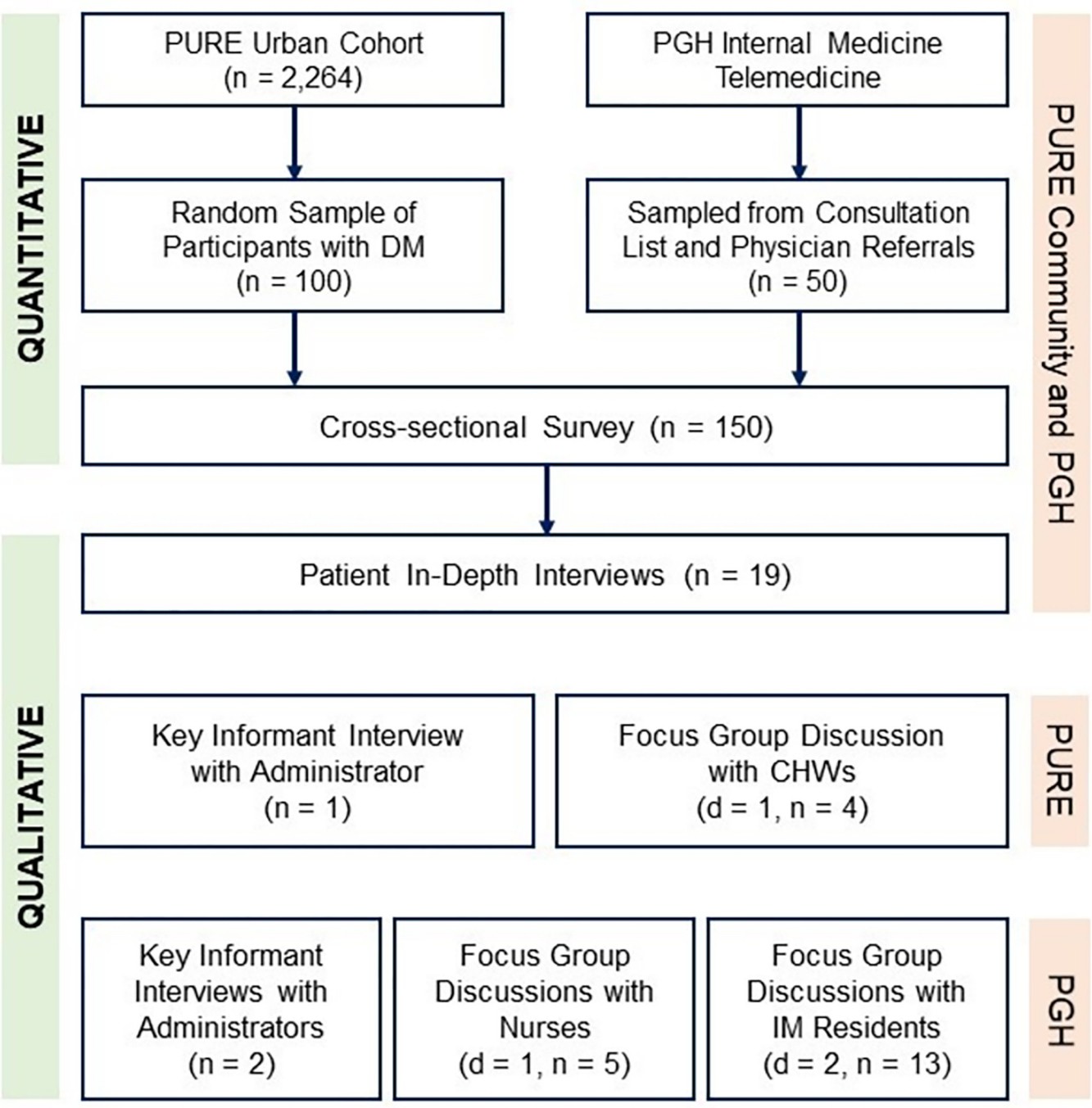

**Fig 1. Study participant flow.** CHW–community health worker, DM–diabetes mellitus, IM–Internal Medicine, PGH–Philippine General Hospital, PURE–Prospective Urban and Rural Epidemiological Study; d–number of focus group discussions conducted, n–number of individual study participants included.

Most participants also reported either foregoing necessities or borrowing money to afford consultations, medications, and diagnostic testing, or having no consultation/meds/tests during the pandemic (Table 4).

**Table 1. Demographic and clinical characteristics of study participants.**

|  | N = 150 (%) |
|---|---|
| Age (years) |  |
| Median [interquartile range] | 62 [55.0, 68.0] |
| Age Groups |  |
| 19–39 years | 3 (2.0) |
| 40–59 years | 54 (36.0) |
| 60–69 years | 58 (38.7) |
| 70 years and above | 35 (23.3) |
| Sex |  |
| Female | 106 (70.7) |
| Education |  |
| Primary | 13 (8.7) |
| High School | 75 (50.0) |
| Vocational School | 11 (7.3) |
| College/University | 51 (34.0) |
| Marital Status |  |
| Never married | 14 (9.3) |
| Currently married | 88 (58.7) |
| Separated/divorced/annulled | 8 (5.3) |
| Widowed | 40 (26.7) |
| Duration of Diabetes Mellitus (years) |  |
| Median [interquartile range] | 10.0 [8.0, 15.0] |
| Number of Any Comorbid Illness |  |
| Median [interquartile range] | 1 [0, 1.0] |
| Types of Comorbidities |  |
| Hypertension | 97 (64.7) |
| Heart Disease | 14 (9.3) |
| Others | 21 (14.0) |

**Table 2. Patient socioeconomic status before and during the pandemic.**

|  | Pre-pandemic N (%) | During Pandemic N (%) | p value |
|---|---|---|---|
| Household Size |  |  |  |
| ≤ 5 members | 99 (66.0) | 102 (68.0) | 0.623 |
| > 5 members | 51 (34.0) | 48 (32.0) |  |
| Presence of Means of Livelihood |  |  |  |
| Yes | 85 (56.7) | 49 (32.7) | <**0.001** |
| Average Monthly Household Income (PHP) |  |  |  |
| <5,000 | 9 (6.0) | 26 (17.3) | **0.017** |
| 5,000–20,000 | 89 (59.3) | 76 (50.7) |  |
| 20,001 and above | 26 (17.3) | 20 (13.3) |  |
| Unknown | 26 (17.3) | 28 (18.7) |  |
| Membership to any Health Insurance Scheme |  |  |  |
| Yes | 124 (82.7) | 123 (82.0) | 0.948 |
| PhilHealth membership |  |  |  |
| Yes | 115 (76.7) | 117 (78.0) | 0.617 |

**Table 3. Characteristics of diabetes care before and during the pandemic.**

| | Pre-pandemic N (%) | During Pandemic N (%) | p value |
|---|---|---|---|
| Duration of Follow-Up in the Healthcare Facility | | | |
| < 1 year | 7 (4.7) | 81 (54.0) | <**0.001** |
| 1 to ≤ 2 years | 23 (15.3) | 24 (16.0) | |
| > 2 years | 120 (80.0) | 42 (28.0) | |
| Frequency of Medication Intake per Week | | | |
| Daily | 129 (86.0) | 119 (79.3) | **0.008** |
| Less than daily | 21 (14.0) | 21 (14.0) | |
| Stopped medications | 0 | 10 (6.7) | |
| Controlled Glucose Level during Last Fasting Blood Sugar Testing | | | |
| Yes | 78 (52.0) | 59 (39.3) | <**0.001** |
| No | 59 (39.3) | 48 (32.0) | |
| Unknown | 13 (8.7) | 9 (6.0) | |
| No recent testing | 0 | 34 (22.7) | |

## Qualitative findings

The qualitative data revealed three overarching themes that influenced their access and the delivery of services for diabetes care during the pandemic–imposition of mandatory lockdowns, financial instability, and *covidization* (i.e., prioritization of COVID-19 above other societal problems) of the healthcare system (Table 5). These were consistent with our quantitative findings.

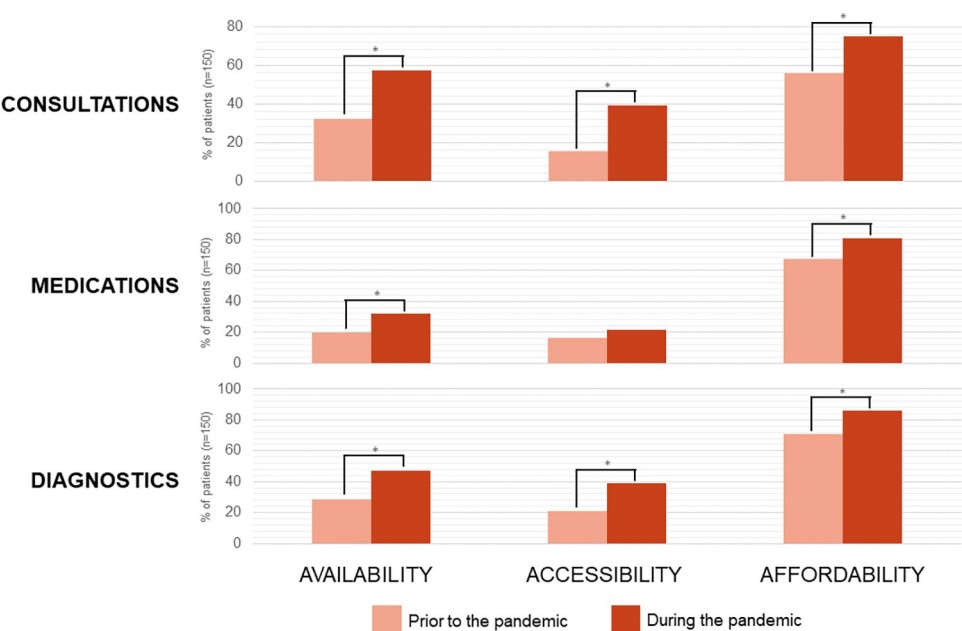

**Fig 2. Patient-reported challenges in accessing diabetes care.** The proportion of patients reporting challenges, defined as being problematic on categorization of Likert scales, in the respective component of diabetes care and domain of access is represented by the y-axis. Asterisks (*) indicate statistically significant differences (p < 0.05) between time periods. There was no correction for multiple comparisons.

**Table 4. Reported catastrophic health expenditure before and during the pandemic.**

|  | Pre-pandemic N (%) | During Pandemic N (%) | p value |
|---|---|---|---|
| Foregoing Other Necessities or Borrowing Money to Afford Consultation |  |  |  |
| Yes | 36 (24.0) | 40 (26.7) | <**0.001** |
| No | 114 (76.0) | 93 (62.0) |  |
| No consult in the last 6 months | 0 | 17 (11.3) |  |
| Foregoing Other Necessities or Borrowing Money to Afford Medications |  |  |  |
| Yes | 40 (26.7) | 50 (33.3) | <**0.001** |
| No | 110 (73.3) | 93 (62.0) |  |
| No medication in the last 6 months | 0 | 7 (4.7) |  |
| Foregoing Other Necessities or Borrowing Money to Afford DM Laboratory Tests |  |  |  |
| Yes | 41 (27.3) | 50 (33.3) | <**0.001** |
| No | 109 (72.7) | 89 (59.3) |  |
| No testing in the last 6 months | 0 | 11 (7.3) |  |

**Imposition of mandatory lockdowns.** The recurrent lockdowns resulted in the suspension of face-to-face consultations and laboratory examinations. Many of the patients in the interview reported having no consultations and examinations done for at least six months, especially during the first two years of the COVID-19 pandemic. Reduced monitoring resulted in uncertainty regarding their diabetes status which they felt compelled to accept. As shared by patient **PG4 (55 years old, female)**:

> When the pandemic started, all those necessary check-ups were totally gone. You know I didn't really know how my diabetes was doing because I wasn't able to go back (to PGH OPD). Even though I know that check-ups and laboratory exams are important, I can't just go out because it was prohibited. It was very difficult during the pandemic.

**Table 5. Summary of qualitative findings.**

| Themes | Key Findings | Data Source |
|---|---|---|
| Imposition of mandatory lockdowns | 1. Limited availability of facilities where patients could seek consultations and laboratory examinations affected patients' capacity to monitor their diabetes.<br>2. Reduced social mobility and lack of public transportation options hindered access to health facilities which had impact on patients' capacity to monitor their diabetes and sustain medication. | Patients |
| Financial instability | 1. Losing sources of livelihood during the pandemic tremendously affected patients' capacity to afford consultations, laboratory exams, and medications.<br>2. Trade-offs were present in the form of other competing household expenses which led to suboptimal management for diabetes.<br>3. Limited capacity to afford medication resulted to different coping strategies such as shifting to herbal remedies, taking intentional medication breaks and utilizing social capital to borrow funds for medicine. | Patients |
| *Covidization* of the healthcare system | 1. The outpatient department, where diabetes was routinely managed, had to temporarily shut down their services for almost a year, effectively restricting access to non-COVID-19 healthcare needs.<br>2. Even with telemedicine, a full assessment for DM patients was difficult to perform without the benefit of physical examination. Additionally, while telemedicine was able to revive provision of outpatient services including diabetes care, healthcare providers felt that many patients were still lost to follow-up due to disparate access to mobile devices.<br>3. Most of the human resources and health supplies were diverted to critical care units to prioritize those with COVID-19. This resulted in delays and suboptimal treatment for admitted patients with complications or severe symptoms from uncontrolled glucose levels. | Healthcare providers, administrators, and patients |

Public transportation became limited as well. Rules on travel were stringent within Metro Manila and even more so for those coming from outside of the region. Despite the eventual ease on heavy lockdowns, travel constraints also had negative impact in accessing facilities for diabetes care.

The restrictions on social mobility coupled with the fear of contracting COVID-19 also impeded some of the patients' access to diabetes care. This was revealed in the account of patient **PG3 (59 years old, female)**:

> *For almost two years, I didn't have any consultations and laboratory exams, but I was taking my medicine continuously. It was the only thing I could do. We were not allowed to go out especially us the elderlies. But I am also afraid to go out even when the most restrictive lockdown was lifted. I still decided not to get any laboratory exams because I don't know, I might get COVID in the diagnostic facility.*

Different experiences regarding access to medications were reported. Some recounted not having any challenges with getting their diabetes medications because pharmacies were still accessible even during lockdowns. While for others, the imposition of granular (i.e., small-scale) lockdowns in certain households precluded any social movement and imposed barriers to medication adherence for certain periods of time.

**Financial instability.** Many of the patients reported losing their source of livelihood during the pandemic. Such experience was not exclusive to the patients themselves but was also a problem for their family members and relatives who often helped finance diabetes care expenditures. Financial instability greatly affected patients' capacity to afford consultations, laboratory exams, and medications. This is exemplified in the succeeding quote:

> **PU10 (50 years old, male):** *My last check-up and laboratory was back in 2019. When the pandemic started, the company I worked for had to lay-off some employees, and I was included. I had a Medicard then* (health insurance from work) *which I used to finance my health expenses for diabetes. Right now I am just continuing the Metformin. . . I'm still not sure when can I have a check-up again because money is an issue. Also, right now you can't just go to the facilities because they require swab tests, that's another thing to be concerned about because you to pay for it.*

Meeting the financial demands for diabetes care resulted in trade-offs. Health expenses competed with other household expenses leading to suboptimal self-management for diabetes as shown in the illustrative quotes below:

> **PU16 (73 years old, female):** *To tell you the truth money was really an issue. My two sons who are living with me lost their jobs, but I am very fortunate that my grandchildren are now paying for my laboratory and medicine. However, money is still tight. Sometimes I buy medicine that's insufficient for a week because we had to pay for water and electricity bills.*

> **PG7 (45 years old, male):** *It was difficult financially. I had problems securing money for my laboratory exams and medications. My work days weren't as frequent as before and my wife lost her job. Plus, we still had other bills to pay.*

Difficulties in purchasing medications were met with different coping strategies intended to manage diabetes within the best of patients' capabilities. These actions include shifting to herbal remedies and taking intentional medication breaks. Some also utilized their social capital by borrowing money from friends and relatives. The FGD with CHWs from the PURE

community also revealed they had to adapt to meet the needs of DM patients within their catchment areas, who mainly relied on free diabetes medication from the local health centers (e.g., house-to-house visit to personally deliver medication to DM patients).

*Covidization* **of healthcare system.** The *covidization* of healthcare has been previously described to illustrate how pandemic-related policies dominated and restructured health service delivery and health research [34–36]. It alludes to the outpouring of resources and attention solely to COVID-19 thereby deprioritizing other serious and pressing health concerns such as the management for chronic conditions.

As revealed in the FGDs with HCWs and administrators, the unavailability of non-COVID healthcare services was a major dilemma for continued diabetes care. Resident doctors from PGH were consistent in reporting that many of their DM patients were unchecked or lost to follow-up during the first year of the pandemic primarily because the OPD had to discontinue services. Many had no contact with their usual DM patients. To some extent the introduction of telemedicine reinstated the delivery of care to OPD patients including those with diabetes. However, healthcare workers still estimated that they were seeing far fewer DM patients than they used to pre-pandemic.

Despite the influx of DM patients who were using teleconsultations, many were still unable to monitor their glucose levels due to social restrictions and lack of funds. Resident doctors agreed that providing medication instructions was one of the major challenges in using telemedicine. Without the benefit of the physical examination, it was more difficult to fully assess the diabetes condition of many patients, especially those who had gone without laboratory testing or had no access to capillary blood glucose monitoring. Instructing patients on insulin use was another struggle for both healthcare providers and patients. Physicians were seeing much more hesitancy among patients towards starting insulin after receiving initial instructions through teleconsultations.

Telemedicine became instrumental in bridging access to diabetes care, but it came with its own challenges. On a micro level, this included inadequate funding to support the sustained operation of telemedicine within the hospital according to administrators. Inequalities regarding access to mobile devices and disparities on technological literacy were also common challenges identified. Hence, while telemedicine was able to revive provision of outpatient services, healthcare workers and administrators felt that many patients were still lost to follow-up due to COVID-19 restrictions.

The FGDs also revealed that most of the human resources and health supplies were diverted to critical care units to prioritize COVID-19 patients. This resulted in delays and suboptimal treatment for inpatients with complications or severe symptoms from uncontrolled glucose levels.

Many patients were also forced to move from one health facility to another to find one that would cater to their non-COVID related concerns. This scenario is described more vividly by patient **PU14 (female, 56 years old)** when she needed emergency care:

> *"There was a time when I was very ill, my family had to send me to a hospital. Turned out my blood sugar was very low. We went to three different public hospitals first, and they all turned us down because they were focused on COVID. We had no choice but to find an alternative that was a private facility where I can get accommodated but we had to pay. That wasn't a pleasant experience."*

Meanwhile, the community doctor and staff members in one of the PURE study sites mentioned having insufficient supply of medication for diabetes and hypertension and further delays in restocking these medicines because of the periodic lockdowns. Additionally, diabetes

care services were limited to dispensing medication. FBS monitoring service was suspended, and the health center staff also took on new COVID-19-related responsibilities such as contact tracing, rapid testing, vaccination, and facilitating of cremation services.

## Triangulating findings

Overall, we found a strong congruence between the quantitative and qualitative findings. The reported limited availability of diabetes care during the pandemic were largely attributed to the *covidization* of the healthcare system. This was especially felt in Metro Manila, which was the region most severely affected by COVID-19 in the country. The reordering of priorities of health facilities and their human resources to COVID-19 activities may have led some patients to discontinue follow-up with their usual diabetes-care provider and seek care in other facilities. These changes may have contributed to the reported increase in the proportion of patients with <1 year follow-up in their most current healthcare facility.

Similarly, the limitations in the accessibility of consultations and laboratory examinations emerged from the imposition of mandatory lockdowns. In the cross-sectional survey, accessibility to medications did not significantly change during the pandemic. However, as shown by the interviews, some patients who experienced granular lockdowns could not reach drug stores, limiting medication adherence. This was reflected in the reduction in daily medication intake of participants during the pandemic.

Finally, the financial instability experienced by patients and their families meant that it was often difficult to afford appropriate diabetes care. Due to loss of livelihood by patients and/or their caregivers, trade-offs including self-management of diabetes were made, often without professional guidance.

## Discussion

We aimed to describe the impact of the COVID-19 pandemic on diabetes care in Metro Manila, primarily from the vantage point of patients, as corroborated by healthcare workers and administrators.

We found adverse changes in the socioeconomic status and diabetes care of patients during the pandemic, most evident in greater proportions of individuals in the lowest income group (i.e., PHP 5,000), which was well below the poverty threshold of PHP 12,030 per month for a family of five [37]. Similarly, that a greater proportion of patients had less than a year of follow up in their current facility during the pandemic suggests that they may have felt compelled to change their facility for consultations. The alarming reduction in the number of patients able to perform FBS monitoring also shows disruptions in a key component of diabetes care [17].

The challenges encountered by patients in accessing diabetes care prior to the pandemic were aggravated during the pandemic. Across all domains and areas of diabetes care, more patients reported challenges during the pandemic, but the accessibility of diabetes medications had not significantly changed, likely due to the continued operations of pharmacies during lockdowns. Despite high enrollment (>75%) into PhilHealth and other health insurance schemes, and service provision being free at point of care in PGH and health centers, most of the patients found challenges in the affordability of consultations, diagnostics, and medications. These difficulties were reflected in our finding that about a quarter of patients experienced catastrophic health expenditures even prior to 2020. The affordability of diagnostics during the pandemic was especially problematic. Our findings echo the well-documented dissociation between PhilHealth enrollment and catastrophic health expenditure [38–40].

Studies on the collateral adverse impact of COVID-19 on healthcare in the Philippines are limited. A qualitative study of public health workers, university staff, and administrators

explored the impact the pandemic on local health systems and higher education institutions in the country. Routine health programs were found to be inaccessible due to changing priorities and lack of students to augment the health workforce [41]. The results were consistent with previous research in other less-resourced contexts [42,43]. Despite suggesting patient hesitancy to access health facilities as a challenge, these studies did not include the perspectives of healthcare users.

Our study is the first to highlight the patient perspective in assessing the indirect effects of COVID-19 on other health programs in the country. We broke down health service access and delivery into availability, accessibility, and affordability of consultations, diagnostics, and medications to more readily identify areas that can be prioritized for improvement. The use of mixed methods allowed us to better characterize the patient experience beyond the constraints imposed by our deconstruction and conceptualization of "access". Furthermore, the perspectives of healthcare workers and administrators helped identify challenges in the local health centers and those associated with the use of telemedicine in the hospital.

Our work adds evidence from the Philippines, which implemented one of the most restrictive responses to the COVID-19 pandemic, about the indirect impact of global pandemics on health. The probability of another extreme novel pandemic occurring within a lifetime is high [44], and as governments and health systems reckon with the aftermath of COVID-19, our work echoes the need to integrate NCD care into pandemic preparedness efforts. Our work also shows how the government's securitization of health and its response to the pandemic may have exacerbated already existing health inequities and highlights the importance of prioritizing health financing strategies in the country to reduce catastrophic health expenditure and to make healthcare truly accessible to all.

Our study has several limitations. First, the relatively small sample size used in the cross-sectional study was only powered to estimate the proportion of the population not receiving DM care and not necessarily the difference in outcomes between two time points. Furthermore, we did not adjust our statistical tests for multiple comparisons. However, even with the limited sample size, the results of the quantitative component consistently suggest generally worse conditions during the COVID-19 pandemic, across different access domains and areas of care.

Second, the catchment population of PGH and the urban location of the community cohort limit the generalizability of our quantitative results to other areas in the Philippines. On the one hand, the longer and stricter lockdowns in Metro Manila would seem to impose greater accessibility issues in the region compared to those outside. On the other hand, the provinces and geographically isolated and disadvantaged areas also experienced lockdowns and resource constraints, and patients may have had even greater accessibility issues compared to those in Metro Manila.

Third, our sampling strategy may have introduced selection bias. By including PGH OPD patients and community-dwelling participants already involved in a different study, the study participants were more likely to already have had better access to healthcare and other positive health-reinforcing behaviors than the average patient with DM. Furthermore, because we excluded patients managed by the diabetes specialty clinic who we assumed to have greater DM care needs, our results are more applicable to assess issues in the primary care setting.

Fourth, inherent in our survey questionnaire design is the potential for recall bias, especially regarding the circumstances prior to the pandemic, and recency bias concerning the events during the pandemic. Study participants who may have experienced negative changes during the pandemic may be more susceptible to viewing events prior to the pandemic in a more positive light. Social desirability bias may have also discouraged participants to admit borrowing money to afford consultation, medications, and laboratory tests; thus, this may underestimate

actual catastrophic health expenditure. To mitigate these issues, the researchers were trained to remind participants to volunteer information based on the operational definitions of 'pre-pandemic' and 'during the pandemic'. For participants recruited from the PGH OPD, the medical records were reviewed to corroborate the answers to the survey. The mixed methods approach also provided opportunities to triangulate the findings from the cross-sectional survey.

Lastly, we iteratively implemented the explanatory, sequential typology such that the cross-sectional study results were not used to inform the construction of the qualitative study guides. Rather, we used the quantitative results to partly inform the analysis of the qualitative portions of the study. We deemed this appropriate in validating our key findings from the patient perspective as corroborated by insights from healthcare workers and administrations.

## Conclusion

The imposition of lockdowns, *covidization* of the healthcare system, and financial instability contributed to the reduced availability, accessibility, and affordability of diabetes care-related consultations, laboratory tests, and medications during the COVID-19 pandemic. These combined effects of the pandemic prompted some patients to self-manage their condition, often without professional guidance. Telemedicine was viewed by healthcare workers and administrators as a promising yet incomplete bridge to reach both new patients and patients lost to follow-up. Our findings highlight the need to integrate NCD care in pandemic preparedness efforts. Our study supports calls to prioritize health financing reforms in the Philippines to prevent catastrophic health expenditure, especially for primary healthcare conditions necessitating routine outpatient care. The reforms are urgently needed now and in preparation for the next pandemic.

## Supporting information

**S1 Checklist. Strengthening the Reporting of Observational Studies in Epidemiology (STROBE) checklist of items that should be included in reports of cross-sectional studies.**
(PDF)

**S2 Checklist. Standard for Reporting Qualitative Research (SRQR) checklist of items that should be included in reports of qualitative studies.**
(PDF)

**S1 Appendix. Likert scales used in the quantitative component of the mixed method study.**
(PDF)

**S2 Appendix. Interview and focus group discussion guides used in the qualitative component of the mixed method study.**
(PDF)

**S1 Data. Cross-sectional survey data.**
(CSV)

**S2 Data. Summary of codes from the qualitative thematic analysis.**
(PDF)

## Acknowledgments

We thank Dr. Jhoanna Rose Velasquez for contributing to the conceptualization and ethics approval application for the study. We acknowledge the assistance provided by Chareece Mae

Zamora-Pudol, Louie Pudol, and Rachel Cabrega in conducting the cross-sectional study. We also acknowledge the assistance provided by Eunice Mallari in conducting and analyzing the qualitative study.

## Author Contributions

**Conceptualization:** Elenore Judy B. Uy, Lijing L. Yan, Truls Østbye, Lia Palileo-Villanueva.

**Data curation:** Greco Mark B. Malijan, John Jefferson V. Besa, Jhaki Mendoza.

**Formal analysis:** Greco Mark B. Malijan, Jhaki Mendoza.

**Funding acquisition:** Elenore Judy B. Uy, Lijing L. Yan, Truls Østbye, Lia Palileo-Villanueva.

**Investigation:** John Jefferson V. Besa, Jhaki Mendoza.

**Methodology:** John Jefferson V. Besa, Elenore Judy B. Uy, Lia Palileo-Villanueva.

**Project administration:** Elenore Judy B. Uy, Lia Palileo-Villanueva.

**Resources:** Lijing L. Yan, Truls Østbye, Lia Palileo-Villanueva.

**Supervision:** Elenore Judy B. Uy, Lijing L. Yan, Truls Østbye, Lia Palileo-Villanueva.

**Validation:** John Jefferson V. Besa, Jhaki Mendoza, Elenore Judy B. Uy, Lia Palileo-Villanueva.

**Visualization:** Greco Mark B. Malijan.

**Writing – original draft:** Greco Mark B. Malijan, John Jefferson V. Besa, Jhaki Mendoza.

**Writing – review & editing:** Greco Mark B. Malijan, John Jefferson V. Besa, Jhaki Mendoza, Elenore Judy B. Uy, Lijing L. Yan, Truls Østbye, Lia Palileo-Villanueva.

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
