## [Decision Letter · Decision Letter 0]

10 Jul 2023

PGPH-D-23-00943

Assessing COVID-19 Pandemic’s Impact on Essential Diabetes Care in Manila, the Philippines: a Mixed Methods Study

Dear Dr. Yan,

Thank you for submitting your manuscript to PLOS Global Public Health. After careful consideration, we feel that it has merit but does not fully meet PLOS Global Public Health’s publication criteria as it currently stands. Therefore, we invite you to submit a revised version of the manuscript that addresses the points raised during the review process.

We look forward to receiving your revised manuscript.

Kind regards,

Nicola L. Hawley

Academic Editor

Journal Requirements:

1. In the ethics statement in the Methods, you have specified that verbal consent was obtained. Please provide additional details regarding how this consent was documented and witnessed, and state whether this was approved by the IRB

2. We ask that a manuscript source file is provided at Revision. Please upload your manuscript file as a .doc, .docx, .rtf or .tex.

3. Please provide separate figure files in .tif or .eps format.

Additional Editor Comments (if provided):

Reviewers' comments:

Reviewer's Responses to Questions

**Comments to the Author**

1. Does this manuscript meet PLOS Global Public Health’s publication criteria? Is the manuscript technically sound, and do the data support the conclusions? The manuscript must describe methodologically and ethically rigorous research with conclusions that are appropriately drawn based on the data presented.

Reviewer #1: Yes

Reviewer #2: Yes

2. Has the statistical analysis been performed appropriately and rigorously?

Reviewer #1: Yes

Reviewer #2: Yes

3. Have the authors made all data underlying the findings in their manuscript fully available (please refer to the Data Availability Statement at the start of the manuscript PDF file)?

Reviewer #1: Yes

Reviewer #2: Yes

4. Is the manuscript presented in an intelligible fashion and written in standard English?

Reviewer #1: Yes

Reviewer #2: Yes

5. Review Comments to the Author

Reviewer #1: In this manuscript the authors provide an important mixed methods examination of how the COVID-19 pandemic (and consequent lockdowns and disruptions to health services) impacted diabetes care experiences among adults living in the Philippines. The authors found that the covidization of the healthcare system, and financial instability was associated with reduced access, affordability, and availability of care. The authors advocate for further and improved integration of NCD care into pandemic preparedness programs. The authors used an explanatory sequential design to provide a mixed methods examination of these topics; participants included diabetes patients, healthcare workers, and health administrators. Methods included semi-structured interviews, focus group discussions, and questionnaires. The analytical plan for the quantitative, qualitative, and triangulation of results was found to be appropriate and sound for the study design and collected data.

Furthermore, the researchers have constructed a well-designed mixed methods study which is of great need to the field of global health.

A few minor recommendations are requested before the paper moves forward.

1. Please define 'covidization' earlier in the paper. Presently it is explained well on page 15, lines 44-45; however, since this phrase is still fairly new to the literature and field, it is recommended this definition be placed earlier in the paper upon the first use of the term.

2. To clarify the explanatory sequential mixed methods design, it would be useful to emphasize that the quantitative data was collected/analyzed first and informed the collection and analysis of the qualitative portion. A figure (Similar to the Figure 1 diagram included would be useful in demonstrating/illustrating the analytic process).

3. As written, the paper leads one to presume that much of the analyses were conducted at very similar time points (this presumption is derived from the statement that participants who completed questionnaires were asked presumably directly afterwards if they wanted to participate in an interview). A strict mixed methods researcher may argue this could fall more into the realm of a convergent parallel design (especially with the emphasis on triangulation of the results). However, there is often flexibility, and as you state, the qualitative portions (including IDIs, KIIs, and FGDs), were used to expand, and explain the quantitative findings as per an explanatory approach. However, the researchers may want to briefly mention this as one minor limitation (that the quantitative results may not have been used sequentially to inform the construction/analysis of the qualitative portion).

4. The sampling and recruitment strategies could use further clarification. Though delineated on page 7, it is not clear why some sample sizes were chosen. For example, was there a particular rationale for why only n = 1 of the key informant interviews were conducted from the PURE study?

5. Page 4, lines 24-25; it is unclear what the (Barangays 178 to 186) is referring to.

6. Greater delineation of the sample sizes, and inclusion and exclusion criteria for all participants could be made. For example, on page 7, lines 23 to 27, it is mentioned that "After the quantitative survey, participants were asked about their willingness to participate further in the qualitative component of the study. Those who expressed interest were screened for inclusion into the IDIs based on age distribution, sex difference, and location. The ideal sample size initially identified for the IDIs was 30."

a. What was the rationale for n = 30 as an ideal sample size?

b. What were the screening criteria for inclusion for the IDIs?

7. Can greater information/clarity be provided about the data availability of the PURE study? How did the researchers access these data to sample participants?

8. Page 6: lines 35 – 40; what was the basis for assuming 50% of the individuals would report not having access to diabetes care during the pandemic?

a. Was the assumption of 8% of the cohort having diabetes derived from the included citation?

9. The researchers may want to consider including the interview and/or focus group agendas as supplementary material.

10. Page 7, lines 27-28: Since multiple criteria can be used to determine saturation, it may be useful to include what criteria was used here.

11. Page 7, lines 40-41; It may be useful to see what the values represented. For example, if 10 is very easy, one can presume that 9 may be 'easy', which leads to questions as to why it would have been defined as problematic.

12. While clearly defined in the manuscript, the number of acronyms used is overwhelming in some sections. It is recommended that alternative phrasing be used.

13. The quantitative and qualitative analyses were well summarized; however, the paper reads a quite a bit longer than most found in public health (largely due to the inclusion of the qualitative portion). The inclusion of mixed methods is of great benefit to the field; nevertheless, it may help the readability and accessibility of this type of study to a wider audience if the researchers could reorganize the results and discussion sections to be a bit more concise (there is some repetition about covidization especially within these sections).

Reviewer #2: - Introduction: more context into how COVID impacted/disrupted health services in the Philippines in particular; there is great data from worldwide effects but want more local information, particularly about diabetes (and other NCDs)

- Methodology: It is unclear what convenience sampling refers to and why participants were recruited in this method. Is the population recruited from PGH OPD equal to those recruited from the PURE cohort? Are these populations the same? What was the follow up on patients ?

- Exclusion criteria: why was the endocrinology subspecialty clinic excluded? what method was used to recruit hospital admin, nurses, residents?

- Analysis: Likert scales used should be provided as supplementary information

- Figure 2: what is the y -axis of the graph? what is being defined as a challenge?

- Table 5: did patients not report covidization of the healthcare system?

- Discussion: what constitutes a catastrophic health expenditure? while providing great findings, not getting the author's opinion as to why these changes lead to covidization and greater health burdens; what are the root causes that lead to secondary effects exacerbated by the pandemic?

6. PLOS authors have the option to publish the peer review history of their article (what does this mean?). If published, this will include your full peer review and any attached files.

**Do you want your identity to be public for this peer review?** For information about this choice, including consent withdrawal, please see our Privacy Policy.

Reviewer #1: No

Reviewer #2: No

---

## [Decision Letter · Decision Letter 1]

31 Oct 2023

PGPH-D-23-00943R1

Assessing COVID-19 Pandemic’s Impact on Essential Diabetes Care in Manila, the Philippines: a Mixed Methods Study

Dear Dr. Yan,

Thank you for submitting your manuscript to PLOS Global Public Health. After careful consideration, we feel that it has merit but does not fully meet PLOS Global Public Health’s publication criteria as it currently stands. Therefore, we invite you to submit a revised version of the manuscript that addresses the points raised during the review process.

We look forward to receiving your revised manuscript.

Kind regards,

Rajat Das Gupta, M.D.

Academic Editor

Journal Requirements:

2. Please provide separate figure files in .tif or .eps format only and remove any figures embedded in your manuscript file. Please also ensure all files are under our size limit of 10MB.

Additional Editor Comments (if provided):

Reviewers' comments:

Reviewer's Responses to Questions

**Comments to the Author**

1. If the authors have adequately addressed your comments raised in a previous round of review and you feel that this manuscript is now acceptable for publication, you may indicate that here to bypass the “Comments to the Author” section, enter your conflict of interest statement in the “Confidential to Editor” section, and submit your "Accept" recommendation.

Reviewer #1: All comments have been addressed

Reviewer #3: All comments have been addressed

Reviewer #4: (No Response)

Reviewer #5: (No Response)

Reviewer #6: All comments have been addressed

Reviewer #7: (No Response)

2. Does this manuscript meet PLOS Global Public Health’s publication criteria? Is the manuscript technically sound, and do the data support the conclusions? The manuscript must describe methodologically and ethically rigorous research with conclusions that are appropriately drawn based on the data presented.

Reviewer #1: Yes

Reviewer #3: Partly

Reviewer #4: Yes

Reviewer #5: Partly

Reviewer #6: Partly

Reviewer #7: Partly

3. Has the statistical analysis been performed appropriately and rigorously?

Reviewer #1: Yes

Reviewer #3: Yes

Reviewer #4: Yes

Reviewer #5: Yes

Reviewer #6: Yes

Reviewer #7: I don't know

4. Have the authors made all data underlying the findings in their manuscript fully available (please refer to the Data Availability Statement at the start of the manuscript PDF file)?

Reviewer #1: Yes

Reviewer #3: Yes

Reviewer #4: (No Response)

Reviewer #5: Yes

Reviewer #6: Yes

Reviewer #7: Yes

5. Is the manuscript presented in an intelligible fashion and written in standard English?

Reviewer #1: Yes

Reviewer #3: Yes

Reviewer #4: Yes

Reviewer #5: Yes

Reviewer #6: No

Reviewer #7: Yes

6. Review Comments to the Author

Reviewer #1: The authors have addressed the reviewer's comments well, and comprehensively. The paper provides a mixed methods approach to understanding healthcare systems access among those with diabetes during the COVID 19 pandemic. The paper is well-presented, well-written, and provides important findings that likely have global relevance. Additionally, the authors provide to the scientific community an excellent example of how to use a mixed methods approach (and specifically an explanatory sequential model) to garner and obtain a nuanced understanding of the health issues and healthcare access among this sample of participants. It is recommended that the paper be accepted for publication.

Reviewer #3: The manuscript demonstrates commendable comprehensiveness and structural organization. However, some critical queries remain:

1.Clarifications are needed regarding the rationale behind exclusion criteria such as "Unable to give informed consent" and "Seeking care at a specialty clinic for diabetes care."

2.The recruitment section should provide numerical insights into the distribution of participants across different recruitment methods (PURE study, PGH OPD telemedicine, physician referrals) and their representativeness vis-à-vis the entire population.

3.A rationale for the relevance of "catastrophic health expenditure" to the study's objectives should be expounded.

4.Address the potential sampling bias introduced by including patients from PGH OPD and the community cohort, emphasizing their limited representativeness of the broader diabetes patient population in the Philippines.

5.Thoroughly discuss the limitations associated with recall and social desirability biases, elucidating their potential impact on result validity and the mitigation strategies employed.

6.In the conclusion, consider incorporating a brief discussion of potential policy implications and recommendations to guide future actions in tackling the highlighted challenges.

Reviewer #4: Thank you for the opportunity to review the manuscript. It is nicely done.

Reviewer #5: The study by Malijan et al investigated pandemic’s Impact on essential diabetes care in Manila. They found that in their country pandemic negatively impacted access to essential diabetes care.

Comments and criticisms. A number of similar studies has been published on the subject in 2021-2022 and the focus on the problem has significantly diminished. I do not think that the present paper will be found of interest by most readers. Maybe a local journal could me the right target. Furhermore the authors themselves underlined the weaknesses of the study, of which the most significant is the small sample size: Manila is a city with millions of inhabitants and reporting the interview of single patients, as reported, is of little meaning.

The simple conclusions are expected and most of the manuscript (too long!) is represented by the method section. Also the introduction is unecessarily long.

Reviewer #6: N/A

Reviewer #7: It is a well written article but lacks the information on how the country's policy on containing the COVID virus, or minimalizing the spread was enforced or how the system was functioning during lockdown and entire pandemic session. Self diabetic care at home by glucometer and how many of the small sample size you have were using it? When the whole world suffered in the pandemic, it is no wonder that NCD care was not optimum in Philippines. What would have been better is if the solution to that was also discussed in KII or at least mention the strategies/methods of preparedness before another lockdown hits in Discussion part. It is also worth mentioning how the country is otherwise managing NCDs. Does it have a free distribution of medicines for already diagnosed diabetic or hypertensive patients who fall under the category of earning <5000/monthly PHP. If the country is making a policy to reduce NCDs, how is it planning for it?

7. PLOS authors have the option to publish the peer review history of their article (what does this mean?). If published, this will include your full peer review and any attached files.

**Do you want your identity to be public for this peer review?** For information about this choice, including consent withdrawal, please see our Privacy Policy.

Reviewer #1: No

Reviewer #3: No

Reviewer #4: No

Reviewer #5: No

Reviewer #6: No

Reviewer #7: **Yes: **Sweta Koirala, MBBS, Ph.D

---

## [Decision Letter · Decision Letter 2]

3 Jan 2024

Assessing COVID-19 Pandemic’s Impact on Essential Diabetes Care in Manila, the Philippines: a Mixed Methods Study

PGPH-D-23-00943R2

Dear Dr. Yan,

We are pleased to inform you that your manuscript 'Assessing COVID-19 Pandemic’s Impact on Essential Diabetes Care in Manila, the Philippines: a Mixed Methods Study' has been provisionally accepted for publication in PLOS Global Public Health.

Best regards,

Rajat Das Gupta, M.D.

Academic Editor

Reviewer Comments (if any, and for reference):

Reviewer's Responses to Questions

**Comments to the Author**

1. If the authors have adequately addressed your comments raised in a previous round of review and you feel that this manuscript is now acceptable for publication, you may indicate that here to bypass the “Comments to the Author” section, enter your conflict of interest statement in the “Confidential to Editor” section, and submit your "Accept" recommendation.

Reviewer #1: All comments have been addressed

Reviewer #4: All comments have been addressed

Reviewer #6: All comments have been addressed

2. Does this manuscript meet PLOS Global Public Health’s publication criteria? Is the manuscript technically sound, and do the data support the conclusions? The manuscript must describe methodologically and ethically rigorous research with conclusions that are appropriately drawn based on the data presented.

Reviewer #1: Yes

Reviewer #4: Yes

Reviewer #6: Yes

3. Has the statistical analysis been performed appropriately and rigorously?

Reviewer #1: Yes

Reviewer #4: Yes

Reviewer #6: Yes

4. Have the authors made all data underlying the findings in their manuscript fully available (please refer to the Data Availability Statement at the start of the manuscript PDF file)?

Reviewer #1: Yes

Reviewer #4: Yes

Reviewer #6: Yes

5. Is the manuscript presented in an intelligible fashion and written in standard English?

Reviewer #1: Yes

Reviewer #4: Yes

Reviewer #6: No

6. Review Comments to the Author

Reviewer #1: The paper provides an examination of the Covid-induced challenges faced by patients living with diabetes in the Philippines. The paper is well-written, and the methods are transparent and thoroughly described. Additionally, the authors have done well to address requested revisions accordingly. The use of a mixed-methods approach is an important and innovative way to showcase the quantitative findings and also capture the lived experiences of participants; it's implementation and analysis is well-done and provides a good example of how to use this method effectively in global health research.

Reviewer #4: (No Response)

Reviewer #6: N/A

7. PLOS authors have the option to publish the peer review history of their article (what does this mean?). If published, this will include your full peer review and any attached files.

**Do you want your identity to be public for this peer review?** For information about this choice, including consent withdrawal, please see our Privacy Policy.

Reviewer #1: No

Reviewer #4: No

Reviewer #6: No
